# Compression hosiery to avoid post-thrombotic syndrome (CHAPS) protocol for a randomised controlled trial (ISRCTN73041168)

Ankur Thapar ,[1,2,3] Rebecca Lawton,[1] Laura Burgess,[1] Joseph Shalhoub,[1] Andrew Bradbury,[4] Nicky Cullum,[5] David Epstein,[6] Manjit Gohel,[7] Robert Horne,[8] Beverley J Hunt,[9] John Norrie,[10] A H Davies[1]

For numbered affiliations see end of article.

**Correspondence to**
Ankur Thapar;
a.thapar09@imperial.ac.uk

## ABSTRACT

**Introduction** Up to 50% of patients develop post-thrombotic syndrome (PTS) after an above knee deep vein thrombosis (DVT). The aim of the study was to determine the effect of graduated compression stockings in preventing PTS after DVT.

**Methods and analysis** Pragmatic, UK multicentre randomised trial in adults with first above knee DVT. The standard of care arm is anticoagulation. The intervention arm will receive anticoagulation plus stockings (European class II, 23–32 mm Hg compression) worn for a median of 18 months. The primary endpoint is PTS using the Villalta score. Analysis of this will be through a time to event approach and cumulative incidence at median 6, 12 and 18 months. An ongoing process evaluation will examine factors contributing to adherence to stockings to understand if and how the behavioural interventions were effective.

**Ethics and dissemination** UK research ethics committee approval (reference 19/LO/1585). Dissemination though the charity Thrombosis UK, the Imperial College London website, peer-reviewed publications and international conferences.

**Trial registration number** ISRCTN registration number 73041168.

## Strengths and limitations of this study

► Pragmatic multicentre randomised trial that will inform international practice.
► Stockings are a low cost, widely applicable, safe intervention across high and low resource healthcare systems.
► Assessor blind design.
► Examines behavioural factors affecting adherence.
► No placebo stocking arm due to ease of breaking blinding.

## INTRODUCTION

Deep vein thrombosis (DVT) occurs in 1–2 per 1000 adults in the UK[1] and half will go on to develop lifelong disability from post-thrombotic syndrome (PTS).[2] PTS is defined as 'chronic venous symptoms or signs secondary to deep vein thrombosis'[3] for example, leg pain, oedema and skin changes, progressing in 5% to ulceration. The average age of patients developing PTS is 55 years, meaning that most are of working age.[4] Individuals with PTS have difficulty walking and therefore maintaining employment, and have a level of disability comparable to chronic obstructive pulmonary disease.[5] The pathophysiology of PTS is sustained venous hypertension from venous outflow obstruction and valvular incompetence.[6]

The recent negative results of the ATTRACT trial have refocussed attention on the effectiveness of graduated compression stockings (GCS) in preventing PTS.[7] The UK National Institute for Health and Care Excellence (NICE) and the American College of Chest Physicians recently withdrew their recommendations for the use of GCS in the prevention of PTS based on the results of the SOX trial.[8 9] However, European guidelines still recommend stockings.[10 11]

A recent systematic review examined randomised controlled trials (RCTs) in this area.[12] Three RCTs inclusive of 1177 patients examined the use of GCS providing 30–40 mm Hg compression at the ankle versus either no stocking,[13 14] or a placebo.[4] Follow-up ranged from 2 to 5 years with a primary outcome measure of cumulative incidence of PTS. There was important clinical, methodological and statistical heterogeneity between trials ($I^2$=94%). Key clinical differences were variable inclusion of patients with chronic venous disease, variable baseline rates of PTS and differing anatomy of DVT. Key methodological differences were the use of a placebo stocking versus a no stocking control arm,

**BMJ**

and differing PTS scoring systems.[15] Additionally, adherence varied between 56% and 93%.

The largest placebo-controlled trial showed no difference in the outcome of PTS with the use of stockings. The other two assessor blind trials showed absolute risk reductions of 23% and 39% with the use of stockings. There appeared to be more benefit from the use of GCS in populations with a higher baseline risk of PTS.[12]

Compression hosiery to avoid post-thrombotic syndrome (CHAPS) is a multicentre, pragmatic, assessor blind, RCT of adults with a first above knee DVT, comparing the regular use of a stocking with no stocking in preventing PTS.

PTS comprises a substantial economic burden on health systems, patients and society due to days lost to illness. The cost of three pairs of high-quality GCS per year is around £150, but this expense may be offset to some extent by lower costs elsewhere. Under standard care, around 50% of DVTs result in PTS.[15] Given the high cost of treatment of PTS, especially venous ulcers and the impact on quality of life,[16] the addition of GCS could be a cost-effective addition to standard treatment.

## METHODS AND ANALYSIS

CHAPS is a multicentre, pragmatic, assessor-blind superiority RCT. The trial will follow patients up for a median of 18 months (range 6–30 months). The study commenced on 1 May 2019 and is due to close on 31 January 2023. Please see figure 1 for a Consolidated Standards of Reporting Trials diagram and online supplemental appendix 1 for a Standard Protocol Items: Recommendations for Interventional Trials checklist.

### Eligibility

Table 1 details inclusion and exclusion criteria. Peripheral arterial disease will be screened for using pedal pulse palpation, with ankle brachial pressure index where equivocal.

### Recruitment

Recruitment will be from emergency departments, ambulatory care, DVT, vascular, obstetric or haematology clinics in 11 UK hospitals (see www.ISCTRN.com for details), via the National Institute for Health Research (NIHR) Clinical Research Network and trial nurses. Recruitment of 864 participants is planned over 24 months from both academic and non-academic centres. Informed consent will be obtained in writing (online supplemental appendix 2) by the local study nurse.

### Study arms
#### Standard care

Anticoagulation for a minimum of 3 months (as per NICE recommendations[8]). The type and duration of anticoagulation beyond 3 months will be determined by local guidelines with the expectation that this will be a direct oral anticoagulant for the majority. A placebo stocking arm was not included because of ease of breaking blinding. GCS are not be recommended for treatment of acute leg pain.[17]

### Intervention arm

Anticoagulation plus a standardised below knee compression stocking (European class II, 23–32 mm Hg compression) worn during waking hours until the end of the trial, or until an alternative is required, for example, compression bandaging for venous ulceration. Minor variations such as change in fabric, open or closed toe or thigh length stockings are permissible if they aid adherence.

A number of behavioural aids will be made available to patients in the stocking arm:
► Patient education video at baseline
► Patient and carer refresher session for stocking donning and doffing within 2 weeks
► Free provision of a donning aid if required
► Cotton stocking for summer use

The following participant retention strategies have been employed:[18]
► Travel cost reimbursement
► Weekly text message reminders to wear stockings
► A Facebook support group for stocking wearers
► Next of kin contact for follow-up

Stockings will be fitted and issued by a local research nurse at first visit. Within 2 weeks, there will be a face-to-face or video call refresher session for donning and doffing with the research nurse, patient and carer.

### Randomisation

Participants will undergo 1:1 web-based randomisation by a local research nurse via the Research Electronic Data Capture (REDCap) database hosted at the Edinburgh Clinical Trials Unit.

### Primary effectiveness endpoint

The primary outcome measure is PTS as assessed by the recommended Villalta score at a median of 18 months follow-up.[19] This will be supplemented by a time to onset of PTS model.

### Secondary endpoints
1. Venous ulceration
2. Employment status (change in number of days working from baseline)
3. Quality of life measured using VEINES (Venous Insufficiency Epidemiological and Economic Study)-QoL and EuroQoL EQ5D scales
4. Adherence to stockings and anticoagulants
5. Cost-effectiveness of stocking prescription

### Sample size

The sample size calculation for CHAPS was based on the cumulative incidence of PTS at 18 months in the recent SOX trial.[4] A minimum clinically important difference of a 10% absolute risk reduction in PTS with GCS was chosen multifactorially, based on patient consultation, the degree of behaviour change required by patients and to be less than that found in earlier positive stocking trials (absolute

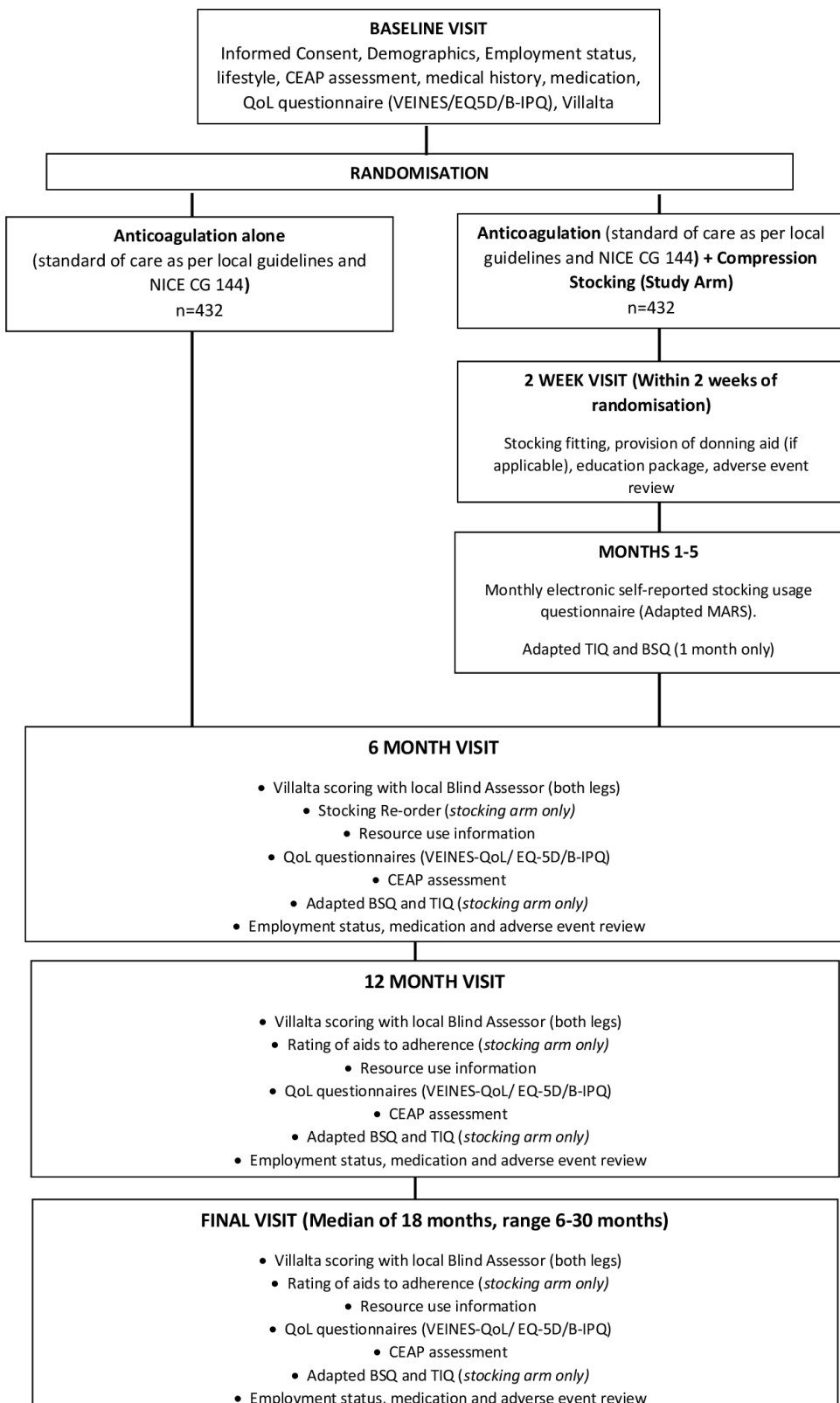

**Figure 1** CONSORT diagram. CONSORT, Consolidated Standards of Reporting Trials; NICE, National Institute for Health and Care Excellence. B-IPQ, Brief illness perception questionnaire; BSQ, Beliefs about stockings questionnaire; MARS, Medicine adherence rating scale; NICE CG 144, National Institute for Health and Clinical Excellence Clinical Guideline 144; QoI, Quality of life; TIQ, Treatment Intrusiveness Questionnaire; VEINS, Venous insufficiency epidemiological and economic study.

| **Table 1** | Eligibility criteria | |
|---|---|
| **Inclusion criteria** | **Exclusion criteria** |
| Symptomatic presentation of first deep vein thrombosis, <2 weeks from diagnosis | Previously intolerant of or already wearing graduated compression stockings for more than 1 month |
| Imaging confirmed, lower limb deep vein thrombosis (popliteal, femoral, iliac or combination) | Contraindication to wearing graduated compression stockings or allergy to fabric |
| Ability to give informed consent | Life expectancy <2 years |
| Age 18 years or over | Ankle brachial pressure index <0.8 (measured when pedal pulses equivocal) |
| | Bilateral deep vein thrombosis |
| | Previous chronic venous insufficiency (patients with existing chronic skin changes or ulceration, defined as C4,5,6 by CEAP classification) |
| | Pre-existing post-thrombotic syndrome, significant leg pain (eg, knee arthritis, spinal claudication) or oedema (eg, lymphoedema) |
| | Newly diagnosed cancer, metastatic cancer or cancer undergoing active treatment or palliation |
| | Contraindication to anticoagulation |

CEAP, Clinical, Etiological, Anatomical, Pathophysiological classification.

risk reduction 23%–39%),[13 14] reflecting improvements in anticoagulation. With 864 participants randomised 1:1, the study will have 90% power at a 5% level of significance using a test of binomial proportions to detect an absolute reduction in the incidence of PTS from 30% in the standard care arm to 20% in the intervention arm, allowing for 10% loss to follow-up. This reduction would represent a number needed to treat of 10 to prevent one case of PTS.

### Internal pilot study

An internal pilot study will follow a randomly selected group of 200 patients over the first 12 months, leading to a mixed-methods process evaluation of factors contributing to GCS adherence.

Adaptations of the Medication Adherence Rating Scale (MARS),[20] Brief Illness Perception Questionnaire (B-IPQ),[21 22] Treatment Intrusiveness Questionnaire (TIQ)[23] and a novel Beliefs about Stockings Questionnaire (BSQ) will be given to participants at the 1 month, 6 and 12 months and final follow-up assessments. Qualitative interviews from a purposive sample of 20 patients at 1 month and 7 months will be used to examine factors affecting GCS adherence in further depth.

### Trial stopping criteria

A combination of self-reported adherence and stocking reordering behaviour will be used to adjudicate adherence at the end of the pilot. The criteria for adequate 1 year adherence is ≥70% of participants in the intervention arm wearing the stocking for ≥4 days per week, with a documented stocking reorder in the last 6 months. This is the remit of the Trial Steering Committee (TSC). If this is achieved, the trial will continue into the main study. If this is not achieved, the trial will terminate and a process evaluation of factors influencing adherence will be reported.

### Assessment of outcomes and of blinding

The study is assessor blind. An independent researcher at each site will perform Villalta assessments blind to treatment allocation. Participants will remove their stockings on the night prior to their clinic visit and be instructed not to discuss stockings during their assessment. The following questionnaires will be administered at follow-up visits: EQ5D, VEINES-QoL and MARS. Employment status (average number of days per week currently working) and healthcare resource use (contacts and outcomes of interactions with health services) will also be collected. Data will be entered by the local research team onto the web-based database REDCap.

Blinding will be evaluated by asking assessors which arm they believe the participant is in. Unblinding is permissible only if a stocking-related significant adverse event is suspected.

### Data monitoring

In line with NIHR recommendations, a Trial Steering Committee (TSC) and an independent Data Monitoring Committee have been appointed to oversee trial conduct (please see online supplemental appendix 3). A Trial Manager together with the TSC will oversee trial progress. The study will be monitored by the Edinburgh Clinical Trials Unit to assess the progress of the study, verify adherence to the protocol and Good Clinical Practice guidelines and to review the completeness, accuracy and consistency of the data, through the use of independent data monitors. Pseudoanonymised data will be stored on REDCap, with a local key held by site Principal Investigators (PIs) to link this to clinical patient records. Data will be filed for 10 years as per local policy and then deleted.

### Data analysis

The primary analysis will be an intention-to-treat analysis that does not adjust for adherence to stockings. This will

be performed independently by the Edinburgh Clinical Trials Unit who will have sole access to the final REDCap data set. This will determine the treatment effect given the adherence in the trial, which is appropriate to gauge real-world effectiveness. The occurrence of PTS will be analysed in both a time-to-PTS approach (since it is possible that the treatment effect may both avert PTS, and also possibly delay its onset) and through analysis of cumulative incidence at a median of 6, 12 and 18 months (as recommended by the International Society for Thrombosis and Haemostasis and peer reviewers).[3] Prespecified subgroup analyses including iliac vein involvement and body mass index >30 kg/m$^2$. To determine the effect of optimum adherence (wearing a stocking for the trial duration for ≥4 days per week) on outcome, a secondary analysis will use Complier Average Causal Estimation modelling through instrumental variable regression. The results of the process evaluation will report which behavioural components change participants knowledge, beliefs and intentions regarding stocking usage. Participants who discontinue the study will have information until date of leaving available for analysis.

### Health economic analysis
Resource arising from the trial interventions, visits and admissions to hospital, general practice visits, community nursing and social and personal care will be collected during follow ups at 6 months, 12 months and the final visit and supplemented by case note review.

Employment status (average number of days worked per week, along with days lost from work and normal activities) will be collected from patients by questionnaire at baseline and at 6 months, 12 months and final follow-up.

A within-trial analysis and a decision model will be constructed. In both cases, the main analyses will be performed from the perspective of the UK NHS and Personal Social Services at 2018/2019 prices. Secondary analyses will be performed from a societal perspective. The results of the analyses will be presented as estimates of mean incremental costs, effects, and, incremental cost per quality-adjusted life year. Sensitivity analyses will be conducted to test the robustness of the results to alternative assumptions about model structure, assumptions and input data. Probabilistic sensitivity analysis will be conducted using Monte-Carlo simulation.

### Ethics and dissemination
The trial was granted ethical approval (National Research Ethics Service ref. 19/LO/1585).

Protocol amendments will be circulated by email to investigators and study nurses to cascade to participants. Dissemination of results will be by the CHAPS coinvestigators in peer-reviewed journals and international conferences and to a lay audience through the Thrombosis UK website.

### Adverse events and liability
All treatment-related adverse events will be collected by site PIs. The chief investigator (CI) will be notified of all serious adverse events within 24 hours. All serious adverse events will be reported to the research ethics committee and sponsor, if, in the opinion of the CI, the event was related to the intervention. All related adverse events and serious adverse events will be recorded and summarised by treatment strategy. The sponsor (Imperial College London) holds a relevant insurance.

### Patient and public involvement
Patients, their carers' and relatives were involved in a three-stage consultation process during the trial development stage, incorporating NIHR INVOLVE methodology. This consisted of a series of semistructured interviews, a survey run via Thrombosis UK, and review of the CHAPS research plan and lay summary. Responses and feedback were incorporated into the design and budget of CHAPS. The Imperial Vascular PPI group has contributed four patients and two members of the public will advise the steering committee for the duration of the trial.

### CONCLUSION
The NIHR funded CHAPS trial will examine whether class II GCS prevent PTS, are cost-effective and the factors influencing adherence at a median 18 months follow-up (range 6–30 months).

**Author affiliations**
[1]Academic Section of Vascular Surgery, Imperial College London, London, UK
[2]Faculty of Health Education, Medicine & Social Care, Anglia Ruskin University, Chelmsford, Essex, UK
[3]Vascular and Endovascular Surgery, Mid and South Essex Hospitals NHS Foundation Trust, Basildon, Essex, UK
[4]Institute of Cardiovascular Sciences, University of Birmingham, Birmingham, UK
[5]School of Nursing, Midwifery and Social Work, The University of Manchester, Manchester, UK
[6]Faculty of Economic and Business Sciences, University of Granada, Granada, Andalucía, Spain
[7]Department of Vascular Surgery, Addenbrooke's Hospital, Cambridge, UK
[8]School of Pharmacy, University College London, London, London, UK
[9]Department of Haematology, Guy's & St Thomas' Foundation Trust, London, UK
[10]Edinburgh Clinical Trials Unit, University of Edinburgh, Edinburgh, UK

**Contributors** AT, JS, NC, MG, BJH developed the protocol and drafted the manuscript. RL and LB developed the PPI section and drafted the manuscript. DE developed the health economic section and drafted the manuscript. RH developed the adherence substudy and drafted the manuscript. JN drafted the manuscript, calculated the sample size calculation and constructed the data analysis plan. AHD is the overall project coordinator and drafted manuscript.

**Funding** This work was supported by the National Institute for Health Research (NIHR), Health Technology Assessment Programme, project number 17/147/47. The HTA Programme is funded by the NIHR, with contributions from the Chief Scientist Office in Scotland and National Institute for Social Care and Health Research in Wales and the Health and Social Care R&D Division, Public Health Agency in Northern Ireland. Sponsor: Imperial College London.

**Disclaimer** The views expressed in this publication are those of the authors and not necessarily those of the NHS, NIHR or the UK Department of Health and Social Care.

**Competing interests** None declared.

**Patient consent for publication** Not required.

**Provenance and peer review** Not commissioned; externally peer reviewed.

**ORCID iD**
Ankur Thapar http://orcid.org/0000-0003-4542-1100

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
