## [Reviewer comments · BMJ Open]

ARTICLE DETAILS

TITLE (PROVISIONAL)	Compression Hosiery to Avoid Post-Thrombotic Syndrome (CHAPS) Protocol for a Randomised Controlled Trial [ISRCTN73041168]
AUTHORS	Thapar, Ankur; Lawton, Rebecca; Burgess, Laura; Shalhoub, Joseph; Bradbury, Andrew; Cullum, Nicky; Epstein, David; Gohel, Manjit; Horne, Robert; Hunt, Beverley; Norrie, John; Davies, AH

VERSION 1 – REVIEW

REVIEWER	Laura Avila The Hospital for Sick Children
REVIEW RETURNED	19-Oct-2020

GENERAL COMMENTS	Thank you for the opportunity to review this protocol. I believe that the use of compression stockings to prevent PTS is a highly relevant clinical question, specially in view of the results of recent trials that investigated the efficacy of thrombolysis in preventing PTS in adults. I have a few minor suggestions and comments. 1) Allocation concealment and implementation of the intervention should be explained in more detail. Similarly, please add details on data collection and data management.2) Will patients who receive thrombolysis for DVT treatment be excluded?3) Experts have highlighted that time from DVT diagnosis to intervention could be critical (ten Cate, Vasa 2016; ten Cate, Blood Reviews 2016). Please detail the study procedures from randomization to intervention. For example, who will measure the patient's leg to obtain the compression sock, who will send it to the compression sock provider, how and when the compression sock will be delivered to the patient, who will teach the patient about sock donning and doffing.4) A pediatric qualitative study has shown education and specialized care to be the most important facilitators for adherence to wearing compression socks. I suggest carefully planning the behavioural aids and considering adding personal support every 3 months to troubleshoot problems with the compression socks and reinforce education. Another point is to consider different approaches for different age groups. For example, some participants may not feel comfortable or have access to a web forum.5) A very important aspect of the trial is the longitudinal analysis of factors that affect adherence. This analysis could provide valuable insight for the development of future strategies to support the intervention. However, the SOX trial (Kahn et al, Lancet 2014) reported a decline in wear over time, with a 96% self-reported use of at one month but 69% at two years. Hence, the timing for the qualitative interviews proposed in the trial (1 and 7 months) may not adequately capture the reasons behind the decline in adherence over time. Please consider adding a third assessment or delaying
---

	the second assessment to the 18-month visit, or even later. 6) A major problem of trials investigating compression socks for PTS prevention, including this trial, is measurement of adherence. As there are no objective ways of assessing adherence at present, I do not think this can be avoided. Hence, every effort should be made to ensure that electronic self-reported compression socks wear is complete. In addition, it would be interesting to explore changes in compression sock pressure at baseline and at 6 months, both in vivo and in vitro, using devices such as the medical stocking tester (Parstch et al, JVS 2006) or dynamometers. These assessments should be performed by independent assessors to protect blinding. The sub-study could provide valuable data to understand compression sock adherence. 7) The use of a binary outcome as end-point is limited when there are other options. I strongly suggest using the Villalta Scale as continuous or ordinal rather than binary. Although the final score is not truly continuous, given that it is obtained by adding ordinal sub-scales, a continuous outcome has better statistical properties for the analysis and will avoid loss of information and increase precision (Altman, BMJ 2006). The authors will need to adjust the analysis to reflect the ordinal or continuous outcome, using mixed models, for example. 8) In addition to using a continuous outcome, I also recommend considering Bayesian methods for trial design and analysis. See https://www.fharrell.com/post/bayes-freq-stmts/. A Bayesian meta-analysis of compression socks to prevent PTS has already shown that there is a large probability of observing some benefit in the population (95% probability of OR <1). The new trial can build upon these reported results. 9) It is likely that 18 months for trial exit is too early to detect potential differences between arms. Please consider extending the study to 24 months. 10) For a study this long, please consider retention strategies (Teague BMC Med Res Meth 2018). 11) Please indicate how employment status and healthcare resource use will be measured and analyzed.
--	---

REVIEWER	Aurelien Delluc University of Ottawa Canada
REVIEW RETURNED	26-Oct-2020

GENERAL COMMENTS	This paper summarizes a randomized controlled trial protocol on use of compression stocking to prevent post thrombotic syndrome in patients with proximal deep vein thrombosis of the lower limbs. This is an interesting topic and challenging study. I would suggest the following prior to considering publication of the paper: The manuscript is hard to follow and is very crude. The different paragraphs of the introduction do not articulate very well. After reading the manuscript, it is not clear to me when the primary endpoint will be assessed for the final analysis. Authors say patients will be followed up for a median of 18 months, but when is the study supposed to end? I do not see any mention of an event-driven study for instance. The authors provided letter of acceptance from NIH to fund the study that included highlighted comments from peer-reviewers. It does not
--

	seem all of these have been addressed in this manuscript. A discussion section is missing. Although it is a protocol, it is common to have a discussion when protocols are published in regular journals. Perhaps feedback from NIH should be addressed here. It is also the occasion for the authors to go more deeply on specific issues. For example, I would be interested in knowing how an MCID of 10% was chosen (and not 5 or 15%), if they have data on how they think they can be successful in enrolling participants, etc. Lastly, in the introduction, authors cite 2018 European guidelines on use of compression stocking to prevent PTS despite recent evidence. I was not aware of these guidelines and checked on Medline: two of the co-authors are employees from SIGVARIS, a compression-stocking manufacturer. I would be very cautious when citing this paper.
--	---

VERSION 1 – AUTHOR RESPONSE

Reviewer: 1

Dr. ML Avila, The Hospital for Sick Children

Comments to the Author:

Thank you for the opportunity to review this protocol. I believe that the use of compression stockings to prevent PTS is a highly relevant clinical question, specially in view of the results of recent trials that investigated the efficacy of thrombolysis in preventing PTS in adults. I have a few minor suggestions and comments.

1) Allocation concealment and implementation of the intervention should be explained in more detail. Similarly, please add details on data collection and data management.

Allocation concealment is discussed on lines 400-402.

The study is assessor blind. An independent researcher at each site will perform Villalta assessments blind to treatment allocation. Participants will remove their stockings on the night prior to their clinic visit and be instructed not to discuss stockings during their assessment.

The following has been added to Intervention section on lines 315-317.

Patients will be measured by their local research nurse who will order the stocking and post it to the patient within 72 hours. A donning aid will be supplied at randomisation if required. Within 2 weeks there will be a face to face or videocall based training session for donning and doffing with the research nurse, patient and carer.

The following has been added at lines 406-407.

Data will be entered by the local research team onto the web based database REDCap.

Data management is dealt with on lines 412-421 in detail. This now includes details of independent data monitors.

2) Will patients who receive thrombolysis for DVT treatment be excluded?

No they are the highest risk for post-thrombotic syndrome and they will be randomised. These patients stand to gain the most if stockings work. Following the results of the ATTRACT trial, we expect this number to be small.

3) Experts have highlighted that time from DVT diagnosis to intervention could be critical (ten Cate, Vasa 2016; ten Cate, Blood Reviews 2016). Please detail the study procedures from randomization to intervention. For example, who will measure the patient's leg to obtain the compression sock, who will send it to the compression sock provider, how and when the compression sock will be delivered to the patient, who will teach the patient about sock donning and doffing.

See additions in lines 315-317

4) A pediatric qualitative study has shown education and specialized care to be the most important facilitators for adherence to wearing compression socks. I suggest carefully planning the behavioural aids and considering adding personal support every 3 months to troubleshoot problems with the compression socks and reinforce education. Another point is to consider different approaches for different age groups. For example, some participants may not feel comfortable or have access to a web forum.

Thank you this will passed on the trial steering committee.

5) A very important aspect of the trial is the longitudinal analysis of factors that affect adherence. This analysis could provide valuable insight for the development of future strategies to support the intervention. However, the SOX trial (Kahn et al, Lancet 2014) reported a decline in wear over time, with a 96% self-reported use of at one month but 69% at two years. Hence, the timing for the qualitative interviews proposed in the trial (1 and 7 months) may not adequately capture the reasons behind the decline in adherence over time. Please consider adding a third assessment or delaying the second assessment to the 18-month visit, or even later.

NIHR have stipulated that if adherence at one year is poor, the factors behind this will be analysed and the trial stopped, as detailed in the section *Internal Pilot Study* lines 370-380. After this point in the main trial at trial close the following qualitative tools will be used to assess factors influencing adherence, as described on 375-380 of *Methods*:

- **Adaptations of the Medication Adherence Report Scale**
- **Brief Illness Perception Questionnaire**
- **Beliefs about Medications Questionnaire**
- **Treatment Intrusiveness Questionnaire**
- **Beliefs about Stocking Questionnaire**

6) A major problem of trials investigating compression socks for PTS prevention, including this trial, is measurement of adherence. As there are no objective ways of assessing adherence at present, I do not think this can be avoided. Hence, every effort should be made to ensure that electronic self-reported compression socks wear is complete.

In addition, it would be interesting to explore changes in compression sock pressure at baseline and at 6 months, both in vivo and in vitro, using devices such as the medical stocking tester (Parstch et al, JVS 2006) or dynamometers. These assessments should be performed by independent assessors to protect blinding. The sub-study could provide valuable data to understand compression sock adherence.

Thank you we do not at present have funding for this substudy but agree it would be interesting.

7) The use of a binary outcome as end-point is limited when there are other options. I strongly suggest **using the Villalta Scale as continuous or ordinal rather than binary**. Although the final score is not truly continuous, given that it is obtained by adding ordinal sub-scales, a continuous outcome has better statistical properties for the analysis and will avoid loss of information and increase precision (Altman, BMJ 2006). The authors will need to adjust the analysis to reflect the ordinal or continuous outcome, using mixed models, for example.

Thank you for this suggestion which we will pass this onto the trials unit and consider it for a secondary analysis.

8) In addition to using a continuous outcome, **I also recommend considering Bayesian methods for trial design and analysis**. See <https://www.fharrell.com/post/bayes-freq-stmts/>. A Bayesian meta-analysis of compression socks to prevent PTS has already shown that there is a large probability of observing some benefit in the population (95% probability of OR <1). The new trial can build upon these reported results.

Thank you for this suggestion which we will pass onto the trials unit and consider it for a secondary analysis.

9) It is likely that 18 months for trial exit is too early to detect potential differences between arms. **Please consider extending the study to 24 months.**

Thank you for this comment. The range of follow up in the trial is 6-30 months. The median follow up is 18 months. This was chosen for several reasons. Firstly, in the positive Brandjes and Prandoni trials, virtually all of the separation between the stocking and control arm occurred by 18 months. Secondly, as the reviewer has pointed out, adherence is poor at 24 months with about 50% of participants discontinuing the intervention. Therefore the applicants believe that 18 months median follow-up is the optimum timepoint to detect a signal

that the intervention works. We agree that if long-term (e.g. 5 year) follow up is required, it can be applied for at this point with additional funding, as per other NIHR trials.

Figure 2: Proportions of patients without symptoms of mild-to-moderate or severe post-thrombotic syndrome

10) For a study this long, please consider retention strategies (Teague BMC Med Res Meth 2018).

Thank you the following has been added to the Method Section (lines 309-313):

“The following participant retention strategies have been employed:

- **Travel cost reimbursement**
- **Weekly text message reminders to wear stockings**
- **A Facebook support group for stocking wearers**
- **Next of kin contact details for follow up”**

11) Please indicate how employment status and healthcare resource use will be measured and analyzed.

The following has been added in the health economic analysis section, lines 451-467:

Resource use required during the trial arising from the trial interventions, visits and admissions to hospital, GP practice visits, community nursing and social and personal care will be collected during follow ups at 6 months, 12 months and the final visit and supplemented by case note review.

Employment status (average number of days worked per week, along with days lost from work and normal activities) will be collected from patients by questionnaire at baseline and 6,12 months and final follow-up.

Reviewer: 2

Dr. Aurelien Delluc, University of Ottawa

Comments to the Author:

This paper summarizes a randomized controlled trial protocol on use of compression stocking to prevent post thrombotic syndrome in patients with proximal deep vein thrombosis of the lower limbs. This is an interesting topic and challenging study.

I would suggest the following prior to considering publication of the paper:

The manuscript is hard to follow and is very crude. The different paragraphs of the introduction do not articulate very well.

After reading the manuscript, it is not clear to me when the primary endpoint will be assessed for the final analysis. Authors say patients will be followed up for a median of 18 months, but when is the study supposed to end? I do not see any mention of an event-driven study for instance.

The following has been added to the Methods section.

The study commenced on 1st May 2019 and is due to close on 31st January 2023.

The following is in the Data analysis section lines 428-431:

The occurrence of PTS will be analysed in both a time-to-PTS approach (since it is possible that the treatment effect may not only avert PTS, but possibly delay its onset) and through analysis of cumulative incidence at 18 months (as recommended by the International Society for Thrombosis and Haemostasis and peer reviewers).

The authors provided letter of acceptance from NIH to fund the study that included highlighted comments from peer-reviewers. It does not seem all of these have been addressed in this manuscript.

All of these points were addressed, the study was subsequently funded by NIHR and has commenced.

A discussion section is missing. Although it is a protocol, it is common to have a discussion when protocols are published in regular journals. Perhaps feedback from NIH should be addressed here. It is also the occasion for the authors to go more deeply on specific issues. For example, I would be interested in knowing how an MCID of 10% was chosen (and not 5 or 15%), if they have data on how they think they can be successful in enrolling participants, etc.

The MCID was chosen multifactorially, as now explained in the *Sample Size* section, lines 320-348.

- **During our patient and public engagement phase, the MCID to justify an investment in behavioural change over 1-2 years was an absolute risk reduction of 10% in PTS.**
- **In previous stocking trials the ARR was 23-39%. With improvements in anticoagulation therapy, we expect this difference to be less**

A Conclusion has been added.

Lastly, in the introduction, authors cite 2018 European guidelines on use of compression stocking to prevent PTS despite recent evidence. I was not aware of these guidelines and checked on Medline: two of the co-authors are employees from SIGVARIS, a compression-stocking manufacturer. I would be very cautious when citing this paper.

Thank you for bringing this to our attention. We agree these guidelines are industry backed and may be biased. However the recent ESVS 2021 DVT guidelines (both referenced in manuscript) also recommend compression stockings.

VERSION 2 – REVIEW

REVIEWER	Laura Avila The Hospital for Sick Children, Canada
REVIEW RETURNED	22-Feb-2021

GENERAL COMMENTS	Thank you for addressing my comments and suggestions.
---

REVIEWER	Aurélien Delluc University of Ottawa / Ottawa Hospital Department of Medicine
REVIEW RETURNED	09-Mar-2021

GENERAL COMMENTS	The authors have addressed my queries.
--